# Endoparasites Infecting Domestic Animals and Spectacled Bears (*Tremarctos ornatus*) in the Rural High Mountains of Colombia

**DOI:** 10.3390/vetsci9100537

**Published:** 2022-09-29

**Authors:** Paula Tatiana Zárate Rodriguez, Luisa Fernanda Collazos-Escobar, Javier Antonio Benavides-Montaño

**Affiliations:** Animal Science Department, Universidad Nacional de Colombia, Palmira Carrera 32 # 12-00, Valle del Cauca, Colombia

**Keywords:** faecal samples, bear parasites, Andean spectacled bear, zoonosis

## Abstract

**Simple Summary:**

The spectacled bear (*Tremarctos ornatus*) is a threatened species, a member of the Ursidae family that lives in the Andes rural high mountain territories of Colombia, Venezuela, Ecuador, Peru, and Bolivia near livestock areas. Parasites in the spectacled bear are a relevant area of interest to preserve this species and understand its habitats and interactions with farm animals. The present work aimed to evaluate the presence of endoparasites in both *T. ornatus* and domestic animals in these areas, by copro- parasitological examination. The results indicate that some parasites have zoonotic potential in wild endangered species and domestic animals in Colombian regions. More sensitive molecular techniques are needed for further identification of the parasite species.

**Abstract:**

This research described the co-infection prevalence of endoparasites in *Tremartus ornatus* and domestic animals in the rural high mountains of Colombia by copro-parasitological examination. Some parasites have a zoonotic potential in wild endangered species and domestic animals in Colombian regions. *T. ornatus* had a notable infection with *Eimeria* spp., *Ascaris* spp., *Ancylostoma* spp., and *Baylisascaris* spp. *Cryptosporidium* spp., *Balantidium coli*, *Anoplocephala* spp., and *Acanthamoeba* spp. In *B. taurus*, *Eimeria* spp. is coinfecting with *Cryptosporidium* spp. (6.6%) and represents 18% of the total parasitism. In *E. caballus* and *B. taurus*. *Eimeria* spp. coinfecting (34.7%), with the *Strongylus* spp. (21.9–25%). In *T. ornatus*, *Eimeria* spp. is coinfecting with *Ancylostoma* spp. (36.2%), *Cryptosporidium* spp., *Ascaris* spp., *Baylisascaris* spp., and *B. coli*.

## 1. Introduction

The spectacled bear (*Tremarctos ornatus*) is a member of the Ursidae family, grouped in three subfamilies: Tremarctinae (spectacled bear, *Tremarctos ornatus*); Ailuropodinae (Giant panda, *Ailuropoda melanoleuca*); and Ursinae (Gray bear, *Ursus arctos*; American black bear, *Ursus americanus*; polar bear, *Ursus maritimus*; Asiatic black bear, *Ursus thibetanus*; sloth bear, *Melursus ursinus*; and malayo, *Helarctos malayanus*) [1,2,3].

*T. ornatus* is a threatened and endangered species according to the Convention on International Trade in Endangered Species of Wild Fauna and Flora (CITES) and the International Union for Conservation of Nature [4]. This species has been systematically studied in its taxonomy, genetics, reproduction, distribution, habitats, diets, behavior, status, and conservation [5] as well as livestock-based conflicts in Colombia, Ecuador, and Bolivia [5,6].

In Venezuela, Colombia, Ecuador, Peru, and Bolivia, Andean bears occupy more than 260,000 km^2^ of forested habitat [7]. These specimens are believed to number over 20,000 adults in these countries [7,8]. Unfortunately, *T. ornatus*’ population has been reduced by 30% to 42% in South America in the last years [9]. These areas are insufficient to guarantee *T. ornatus*’ preservation [7].

Parasites in Andean bears *T. ornatus* are a relevant area of interest. Although there is minimal information about the endoparasites of *T. ornatus* in Colombia, a notable study developed using coprological techniques in the Chingaza National Park described the presence of *Cryptosporidium* spp., *Ascaris* spp., *Baylisascaris* spp., *Microsporidium* spp., *Trichostrongylus* spp., *Strongylus* spp., *Blastocystis* spp., *Fasciola* spp., and *Trichomonas* spp. in *T. ornatus* [10].

In countries such as Peru, in a wildlife refuge at Yanachaga Chemillen National Park, authors reported the apicomplexans *Blastocystis* spp. protozoa, *Cryptosporidium* spp. (14.3%), ciliates such as *Giardia* spp., and three nematodes: *Strongyloides* spp. (25%), an undetermined species of Ascarididae (21.4%), and Ancylostomatidae. The most significant prevalence of parasites belonged to the Strongyloididae family (25%), followed by Ascarididae (21.4%) and Cryptosporidiidae (14.3%) [11].

More parasites in fecal samples have been identified during the dry season (87.5%) than in the rainy season (16.7%). Up to date, eight species of endoparasites and one species of ectoparasites have been identified in Andean bears [11]. The black bear (*Ursus americanus)* is the most researched species in this topic; however, its ecological niches are different from *T. ornatus*’.

Recently, a new parasites species was discovered: *Baylisascaris venezuelensis*. This species is closely related to *Baylisascaris transfuga* [12], a parasite of the giant panda (*Ailuropoda melanoleuca).* This relationship suggests that this panda species could probably be a reference for studying parasites in *T. ornatus* [13]. We consider that *T. ornatus* must be studied rigorously considering its ecological distribution and food habits. Therefore, it requires better biological support to know more about its parasite dynamics. In this study, we report endoparasites in domestic animals and *T. ornatus* at the high altitude of the central Andes, where domestic animals and *T. ornatus* live in common areas. We aim to contribute information about *T. ornatus*’ ecology and parasite niche relations.

We found endoparasites in domestic animals and wild bear populations in high rural areas of Colombia using copro-parasitological methods. Future studies may complement these results through the implementation of biomolecular analyses. Parasites such as *Eimeria* spp. are present in both domestic animals and *T. ornatus*, coinfecting with other parasites such as *Cryptosporidium* spp. and *Buxtonella sulcata*.

## 2. Materials and Methods

### 2.1. Study Area and Population

This field study was carried out in the department of Valle del Cauca, in the rural area of Palmira and Cerrito, in the districts of Combia (lat: 3°40′325″ N, long: 076°03′058″ E, alt: 2179.9 m.a.s.l), Tenerife (lat: 03°44′411″ N, long: 076 04′956″ E, alt: 2898–3844 m.a.s.l), Cañon del Combeima (lat: 04°33′467″ N, long: 075°19′251″ E, alt: 1592–2305 m.a.s.l) during the rainy months of 13 July and 6 December 2021; and the village of Gabriel Lopez, Totoró Municipality, which is located in the Valle de Malvazá, 20 km east of the capital of the Cauca department, 3000 m.a.s.l.

#### 2.1.1. Rural High Mountains of Tenerife, Valle del Cauca

Tenerife is located 1750 to 2750 m.a.s.l, with temperatures of 2 °C–14 °C. The climate is dry, with a relatively well-defined dry season from January to June and a rainy season from July to December. On the other hand, Combia is located in the rural area of Cerrito, where their inhabitants have reported bear attacks. With fewer than ten animals per owner, Combia’s residents have a small production system that guarantees food security through the production of poultry, eggs, milk, and meat [14].

#### 2.1.2. El Silencio, Cañon del Combeima (Tolima)

This is located in the Central Mountain system, within the Parque Nacional Natural los Nevados, on the way to the Nevado del Tolima, at 2600 m.a.s.l. Its waters are essential to sustain the production of Colombian coffee, rice, sorghum, cotton and corn. The wild animals that require preservation in this area are *Tapirus pinchaque* (mountain tapir), *T. ornatus* (spectacled bear), *Pudu mephistophiles* (northern Pudu), *Odocoeilus virginianus* (white-tailed deer), *Silvilagus andinus* (*Andean tapeti*), *Leopardus tigrinus* (oncilla) and *Puma concolor* (puma) [15,16].

#### 2.1.3. The Village of Gabriel Lopez, Municipalities of Totoró (Cauca)

This is located in the Valle de Malvazá, 20 km east of the capital of the department of Cauca, at 3000 m.a.s.l. Its temperatures range between 9 °C and 19 °C. The economic activity of its inhabitants revolves around agricultural products such as potatoes, fique, coffee, and aromatics [17]. It borders the Paramo de las Delicias (central mountain range) and the upper basin of the Cauca River, where several water sources of importance are born, such as the Palace River [18]. There are numerous reports of bears attacking and eating cattle and horses in these areas (Figure 1).

The geographic location. High altitude of the central Andean mountains. The farms are located at the border of *T. ornatus* territory, 2600 to 4100 m.a.s.l. Valle del Cauca, Tolima, and Cauca (Colombia). Generated with ArcGIS, 10.8.1 version of SIG laboratory, Universidad Nacional—Palmira.

### 2.2. Description Area

Most farms in the rural high mountains are centered on the production of beef and dairy cattle. It is a traditional system without technical support in which calves are allowed to graze freely or are stocked and brought in for lactation twice a day. Diarrhea in calves was reported in some farms. Most animals drink water from rivers or small ponds without water treatment (non-potable).

Some areas have small farms with pigs, cattle, horses, sheep, and pets such as dogs and cats. *T. ornatus* transits through livestock lands to find food, for instance, “piñuelas” Puya furfuracea (Willd.). The Valle del Cauca is rich in “frailejones” Ruilopezia cardonae (Cuatrec.), Speletia steyermarkii Cuatrec and Hesperomeles goudotiana “mortiño colorado” [19]. (Figure 2). Some cattle owners move animals to high altitudes for them to graze in *T. ornatus*’ land, invading and affecting this bear’s territory, while also contaminating rivers and water sources (Figure 2A–C).

### 2.3. Type of Study

This cross-sectional study sought to assess the associations between the disease or health-related traits and other variables of interest in a specific population and time. The presence or absence of the disease and its variables were examined in a sample and without considering the temporal sequence of cause and effect [14]. The prevalence of gastrointestinal parasites was estimated using prevalence (*p*) = the number of total cases divided by the sum of the population at the moment (×100). The data were expressed in percentages (%). We used at least three stool samples to accurately diagnose parasitic intestinal infections (IPI) with a 95% confidence interval (CI). The value for significance of the association and allowable error was 0.05 [20]. We collected 58 stool samples from *T. ornatus*, but we estimated the populations of bears to range from 40 to 60 specimens.

### 2.4. Samples

Stool samples (10 g) were obtained from domestic animals, horses, and cattle on the border of the reserve forest, directly from the rectum. Between 13 July and 6 December 2021, we collected fresh feces in the morning (6–12 h old), which were identified with the aid of an experienced park ranger. Fresh samples were recognized by their brown or green color. Saline wet mounts were made by mixing approximately 2 mg of stool with a drop of physiological saline on a microscope glass slide and placing a coverslip over the stool suspension. Samples were also analyzed using iodine wet mounts and microscopically examined with the afore mentioned method. The wet mounts were studied microscopically with a low power objective (10×) followed by switching to a high-powered one (40×). Each stool sample was screened by an experienced microscopist before reporting negative results. Additionally, the Zieh Nielsen technique was employed using 10 g of fuchsine diluted in 100 mL of ethanol and a 5% of phenol solution (5 mL of phenol and 95 mL of water). Then, 10 mL of basic fuchsine was filtered, and 100 mL of phenol solution was added in order to form the mother solution. Excess alcohol was removed with tap water and discolored with 7% H_2_SO_4_ until the plate was pale pink, forming a sulphuric acid solution (7% H_2_SO_4_, 7 mL of sulphuric acid mixed with 93 mL of Ethanol). Excess colorant was also removed with tap water, and then we added methylene blue or malachite green, spreading it for 3 min. 10 g of methylene blue was diluted in 95% ethanol, and then 30 mL was filtered from the 100 mL of the mother solution; afterwards, 70 mL of water was added. The malachite green solution was conformed of 5 g malachite green diluted in 10% ethanol, 100 mL). The excess colorant was eliminated with tap water and left to dry in order to visualize the plate with immersion oil, using the 100× objective. The parasite analysis was performed with direct microscopic examination using a ZEISS AxioCam ICc 1 microscope, with flotation using the Sheather technique and sedimentation methods, as well as fixation and coloring techniques of Zieh Nielsen [10,14]. Samples were stored at −20 °C for future molecular studies.

## 3. Results

From 264 fecal samples collected from domestic animals and *T. ornatus*, we identified that 98/264 specimens were positive to at least one parasite, with a total prevalence of 60.93%. 35/58 were prevalent in *T. ornatus* (60%) [95% CI = 48–73%], 31/112 in *B. taurus* (28%) [95% CI = 8–26%], and 22/48 in *E. caballus* (46%) [95% CI = 26–60%] (Table 1).

Most samples from domestic animals were collected from *B. taurus*, Equidae *Equus caballus*, *Equus asinus* and their crossing. Samples from calf and young bears were more soluble than adults’ feces. None of them had blood, mucus, or clinical parasitic symptoms.

Most of the bear samples were soft with green and brown color due to the nature of the vegetable tissue in the animals’ feeding area (Figure 2E,F). Some samples had a red fruit smell.

We studied the prevalence of parasites associated with more than one species considering the total number of samples (264). We identified that the most frequent association in *B. taurus* was *Eimeria* spp. with *Cryptosporidium* spp. (4/60, 6.6%). *Eimeria* spp. represents 18% of the total parasite associations, followed by *Cryptosporidium* spp. In horses and cattle, Eimeria had a strong association with other parasites (34.7%), but most co-infections were associated with the Strongyle family (21.9–25%). In bears, there was a robust parasite co-infection with *Eimeria* spp., *Ancylostoma* spp., *Cryptosporidium* spp., *Ascaris* spp., *Baylisascaris* spp., and *B. coli* (36.2%) (Figure 3).

Using microscopy techniques, we observed different parasites with morphological characteristics and compatible measures with *Buxtonella sulcata*, *Eimeria bovis*, *Eimeria zuernii*, *Strongylus vulgaris*, *Ascaris lumbricoides*, *Baylisascaris venezuelensis*, *Ancylostoma ailuropodae* and *Eimeria pellita*; however, further studies using molecular techniques are required to confirm these classifications (Figure 4 and Figure 5).

For example, a cyst of *B. sulcata* measuring 61.324 × 60.97 µm (Figure 4A) is compatible with a cyst of *B. sulcata*, which is oval-shaped or round-shaped, yellowish green in color and measuring 54.8–96.2 μm in diameter, with a mean of 67.3 ± 11.1 μm. A double-layered capsule that displays a macronucleus and contractile vacuoles surrounds these cysts (60–68.6 × 60–68.8 µm) [21,22].

In the case of Figure 5F, the 58.9 × 57 µm cyst is consistent with *Balantidium coli*, a smaller and dark cyst, measuring 40 × 60 µm [23]. The findings in *T. ornatus* can be attributed to the small pig production systems at high altitude bordering this bear’s lands. Figure 5B shows an egg with 48.7 × 105 µm in size, compatible with *Strongylus vulgaris* (83–93 × 48–52 µm). Figure 4D displays a cyst measuring 30.9 × 20.5 µm, consistent with *Eimeria bovis* (25–34 × 17–23 µm) [24]. The cyst measures 20.5 × 19.2 µm, which is compatible with *Eimeria zuernii* (15–22 × 13–18 µm) Figure 4F [24]. The egg in Figure 5C measures (51 × 36 µm), which is within the dimensions of *Ascaris lumbricoides* (45–75 × 35–50 μm) [25]. The oval-shape and size of Figure 5D, measuring 63 × 77 µm, is compatible with *Baylisascaris venezuelensis* (66.3–74.7 × 78.3–88 µm) [12].

## 4. Discussion

In this study, 264 faecal samples using coproparasitological examinations techniques help to identify gastrointestinal parasites in *Tremartus ornatus* and domestic animals in the rural high mountains of Colombia. This technique has a lower operational cost and moderate sensitivity and specificity. These techniques are biologically useful, but they need to be complemented with biomolecular technologies in future studies to better understand the biological relations between host and the biology of parasites due to the difficulties in obtaining samples from these animals and optimize the effort in undeveloped countries, where there is limited knowledge available and research investment [14,26,27].

Interestingly, the prevalence of *Eimeria* spp. in *T. ornatus* (30%) in this study is biologically relevant (Table 1). The following parasites have been previously reported in *Ursus americanus*: *Eimeria albertensis* and *Eimeria borealis* [28]. In giant panda: *Ailuropoda melanoleuca*, Eimeria, with a prevalence of 15.9% [26]; in red panda: *Ailurus fulgens*, *Eimeria* spp. (67.44%), which is also the most prevalent parasite [27]. Similar studies report *Eimeria* spp. (47.32%) in Himalayan black bear, *Ursus thibetanus*. Additionally, *Eimeria ursi* has been found in brown bears, *Ursus arctus*, in Eurasia [28]. In Colombia and Ecuador, coccidiosis and *Eimeria* spp. in *T. ornatus* has also been reported, but the specific species have not been identified [10,29].

In our study, we found that *Eimeria* spp. was also the most prevalent (33.08%) in horses. We also found cyst of *Eimeria* spp. (53.89%) in *B. taurus*, which had a similar prevalence to the reports of other studies in low and high altitudes (17.4–77.9%) [30,31]. Parasite species such as *Eimeria* spp. might be transmitted from cattle to bear and vice versa, and probably, as stated previously, the host specificity of this parasite might be caused by adaptive rather than cophylogenetic processes [32,33].

In the case of *Cryptosporidium* spp., *Giardia* spp., and *Microsporidium* spp., our study found traces of them in *B. taurus* (5.36%, 3.57%, 2.68%); *E. caballus* (4.17%, 0%, 1.7%) and *T. ornatus* (10%, 1.7%, 0%). Enteric protozoa such as *Cryptosporidium* spp. and *Giardia* spp. are responsible for causing diarrhea and even death in neonatal and young bovine calves [34,35]. The prevalence reported for cryptosporidiosis in humans, animals, and water sources were 7.8%, 20.4%, and 38.9%, respectively [36].

In horses, we found a 4.2% higher prevalence in this species than in other countries, where the value is 2.3% [37]. We also identified that, in horses, there is association with *Microsporidium* spp. (6.25%). This data is consistent with previous studies [37]. In our research, *Eimeria* spp. was found circulating in *B. taurus*, *E. caballus*, and *T. ornatus*. *Cryptosporidium* spp. is circulating in *B. taurus*, *E. caballus*, and *T. ornatus*. *Microsporidium* spp. is infecting *B. taurus* and *E. caballus*, and finally, *Buxtonella* spp. was identified in *B. taurus*, *E. caballus*, and *T. ornatus* (Figure 6).

Giardia in horse was not detected in this study with coprological techniques, but *G. duodenalis* (17.4%) has been previously reported in Colombia’s horses using PCR [38]. *Giardia* spp. in cattle and *T. ornatus* has been previously reported in domestic animals and wildlife, particularly *G. duodenalis* in livestock [39,40]. This parasite was reported in *T. ornatus* by Figueroa, in Peru [41]. *G. duodenalis* is a common anthropozoonotic parasite [42].

Although this is the first evidence of *Giardia* spp. in both species (*B. taurus* and *T. ornatus*), it is essential to know the level of parasites impacting their health. This information may have consequences for conservation, associated with nutritional stress, parasitism, and the human-cattle-*T. ornatus* conflict. As such, intervention may be needed to prevent further damage [43,44]. Genetic characterization of Giardia isolates from humans and *T. ornatus* and the water used in a closed environment will help to understand the transmission routes and the level of association of this parasites in farms where cattle-horses and bears share common spaces in Colombia at high altitude [45].

The microsporidia are obligate intracellular parasites consisting of at least 200 genera and 1400 species, infecting a broad range of animals (vertebrates and invertebrates). They infect fish, insects, farm animals, humans, and companion pets, leading to zoonotic transmission and affecting immunocompetent and immunocompromised humans [46,47,48]. In giant panda (*Ailuropoda melanoleuca*), *Enterocytozoon bieneusi* has been identified through PCR techniques with a positive rate of 35.5% [49].

*E. bieneusi* is the most common human-infecting microsporidian species, which includes pathogens of diverse companion animals and livestock [50]. Fast evolutionary rates, host switching using distant related hosts and habitats, as well as habitats destruction, environmental stress, extensive animal farming, and human encroachment on wild ecosystems may drive these new host-parasite interactions [50].

*Microsporidium* spp. in *T. ornatus* was not reported in our study, but we encourage further research using more sensitive molecular techniques on biological evidence, considering that *Microsporidios* spp. showed a prevalence of 16.66% in a study developed in the Chingaza National Park [10].

Another cattle–bear–horse parasite prevalence was *B. sulcata*, an opportunistic ciliate protozoan cattle and water buffalo ciliate [51] that inhabits the colon of cattle, causing diarrhea and debilitating the animals. Despite sporadic reports in the literature from the Indian subcontinent [52], it can be misdiagnosed as *B. coli*, a ciliated protozoan found in the cecum and colon of humans, nonhuman primates and pigs [52,53]. In this study *B. coli* was present in horses (4.17%), *T. ornatus* (1.7%), and Cattle (0.89%). Higher infection rates have been reported in cattle (9.9–23.6–38.5%), suggesting the influence of protozoan diarrheal symptoms in bovines [22]. In Egypt, studies conveyed a prevalence of 32.86% [54], 27.7% in Uruguay [55], and 0.32% in *Camelus dromedarius* [54] and 6.25% in Cattle from Colombia [56,57,58]. *Buxtonella* spp. has also been identified in feces of rhesus macaques, hamadryas baboons (*Papio hamadryas*) and agile mangabeys (*Cercopithecus agilis*) [59].

Interestingly, we did not find previous reports of *B. sulcata* in horses. Probably the parasite was introduced to America by the Spanish conquistadors, who obtained their horses in northern Africa, where they had been in contact with camels infected with *Infundibulorium cameli* syn of *B. sulcata.* Future studies are required to test the association with horses [60].

Regarding nematodes identified in *T. ornatus* during this study, we found *Ascaris* spp. (21.7%), *Baylisascaris* spp. (13.33%), *Ancylostoma* spp. (15%) and *Strongylus* spp. (1.67%). Other studies developed in *T. ornatus* reported *Ascaris* spp. (55.55%), *Baylisascaris* spp. (38.88%), *Trichostrongylus* spp. (11.11%) and *Strongylus* spp. (16.66%) in the Chingaza National Park of Colombia [10]. In Ecuador, *T. ornatus* in captivity were found infected by *Ancylostoma* spp. and *Ascaris* spp. [29].

Parasites such as *Baylisascaris* spp. and *Ascaris* spp. have also been reported in *T. ornatus* at the zoological or captivity level in USA [61]. Likewise, they have been identified in fecal samples from wild populations in Venezuela and Peru (Strongyloidea, Ascarididae, and Ancylostomatidae) [41]. During our study, we found associations between *Ascaris* spp.—*Ancylostoma* spp. (3.44%) and *Baylisascaris* spp.—*Ancylostoma* spp. (1.72%) (Figure 3 and Figure 6). In rural high mountains, there have been reports in domestic animals’ nematodes from the Ascarididae family in *Toxocara cati* (44%), *Toxocara canis* (25%), and *Parascaris equorum* (37%) [14].

*Baylisascaris* spp. has been previously published as *B. venezuelensis*, since it has already been characterized in *T. ornatus* using molecular techniques and compared with *Baylisascaris transfuga*, which has shown a 52.9% prevalence in brown bears [12,62]. 

*Baylisascaris* spp. has a monoxenous life cycle [63] and high potential to cause visceral, ocular, and neural migratory larvae in a range of different hosts, such as mammals and birds; therefore, they represent a zoonotic risk [62]. It is critical to warn tourists to prevent a zoonotic outbreak, considering that *B. procyonis*, *B. columnaris* and *B. transfuga* are described as etiological agents of migratory larvae [12,61]. Regarding *B. venezuelensis*, its level of pathogenicity in bears it is unknown, even though *B. schroederi* in pandas is a significant cause of morbidity and mortality. Additional research on the potential risk of *B. venezuelensis* to spectacled bears is needed [64].

Regarding other nematodes, we report *Ancylostoma* spp. (15%) in *T. ornatus*. This species has also been found in Colombia with a prevalence of 5.55% in *T. ornatus* [10]. *Uncinaria* sp. has also been documented in the American black bear, *Ursus americanus*, brown bears and polar bears, *Ursus maritimus* [65].

Regarding *Strongylus* spp., we found a prevalence of 1.72% (1/58), which is less than that reported in *T. ornatus* (16.67%) in Chingaza, Colombia in [10]. Similarly, a prevalence of 25% was reported in Peru [41].

The interaction or multiple associations between wild animals and domestic animals and humans are not completely understood [66], and the potential role of hosts for transmission of zoonotic diseases in rural high mountains is not completely explored, as well as other wild animals that can trigger different dynamics. Zoonotic parasites such as *Uncinaria* spp., *Strongyloides* spp., *Baylisascaris* spp. and *Cryptosporidium* spp. are present in *T. ornatus* and domestic animals. This environment can cause potential larval migrans, skin problems as well as enteric human, domestic and wild infections.

Previous reports in humans at high rural mountains by Peña-Quistial shows that *Toxocara canis* and *Toxocara cati* had a prevalence of 24% and 44% [14] indicating that these parasites might be circulating in domestic animals that are able to cause larva migrans [14,66]. In the case of *Baylisascaris* (*Ascarididae* family, *Ascaridida* order, phylum Nematoda), its potential role to infect other animals as well as the agent that can cause larva migrans in humans and animals requires further research.

Finally, further research is needed to better understand parasitic dynamics in different seasons and the parasites’ effects on these populations in the high rural mountains of Colombia, where farms located at this altitude increase the likelihood that the mountain bear *T. ornatus*, under low food conditions, extreme climate events, and deforestation and fragmentation processes, is forced to increase its interaction with domestic animals, which will continue to drive human–bear conflicts [11,67].

## 5. Conclusions

Endoparasites such as *Eimeria* spp. in *T. ornatus* and *Cryptosporidium* spp. and *Buxtonella sulcata* are common parasites in *T. ornatus*, *B. taurus* and *E. caballus* that require further studies around the clinical effects in these populations.

We recommend developing seasonal parasites studies as well as research regarding the population dynamic of each parasite to know the levels of exposition throughout the year. Future studies are also needed to identify other parasites species association among *T. ornatus*, wild and domestic animals.

## Figures and Tables

**Figure 1 vetsci-09-00537-f001:**
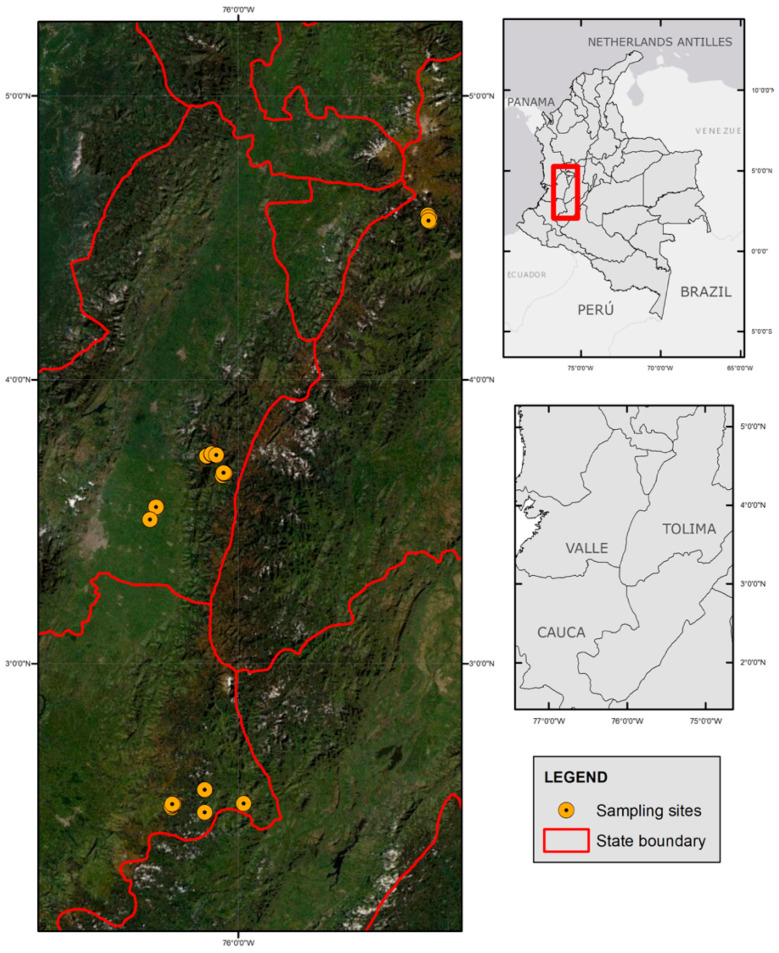
The geographic location. High altitude of Central Andean Mountains. The farms are located at the border of *T. ornatus* territory, 2600 to 4100 m.a.s.l. Valle del Cauca, Tolima, and Cauca (Colombia). Generated with ArcGIS, version 10.8.1 of SIG laboratory, Universidad Nacional—Palmira.

**Figure 2 vetsci-09-00537-f002:**
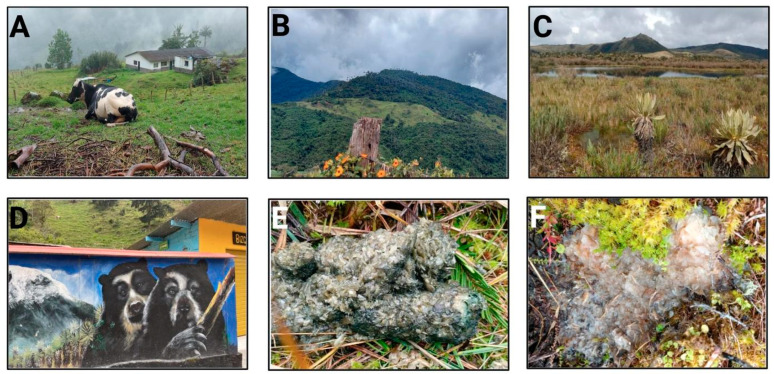
Cattle farms at the High altitude of Central Andean Mountain are located at the border of *T. ornatus*’ territory (**A**–**C**). Awareness-raising campaign to conserve the land of the Andean bear and protect its territories (**D**), Feces collected from *T. ornatus* (**E**,**F**).

**Figure 3 vetsci-09-00537-f003:**
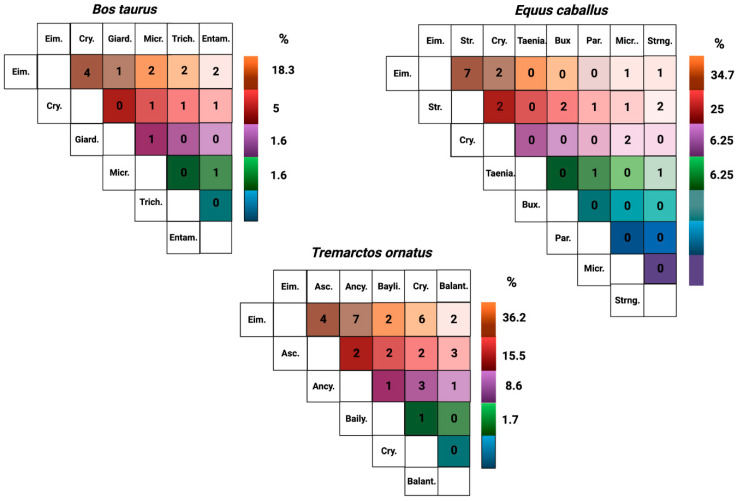
Quantity of individuals for each possible combination of endoparasites. Eim: Eimeria, Cry: *Cryptosporidium* spp., Giard: *Giardia* spp., Micro: *Microsporidium* spp., Trich: *Trichostrongylus* spp., Entam: *Entamoeba* spp., Str. *Strongylus* spp; Taenia: *Taenia* spp., Bux: *Buxtonella sulcata*, Par: *Parascaris equorum*, Std: *Strongyloides* spp., Asc: *Ascaris lumbricoides*, Ancy: *Ancylostoma* spp., Bayli: *Bailisascaris venezuelensis*, Balant: *B. coli*.

**Figure 4 vetsci-09-00537-f004:**
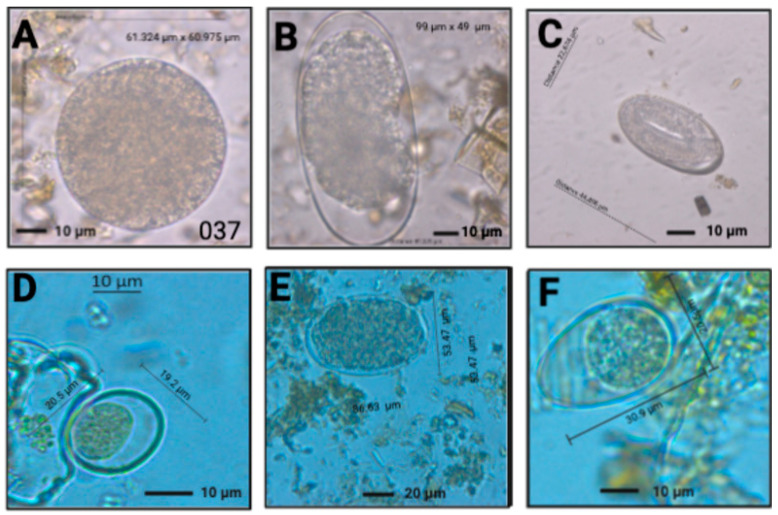
Eggs and cysts found in *E. caballus* and *B. taurus*. *E. caballus*: *Buxtonella sulcata* (**A**), *Strongylus* spp. (**B**) *Strongyloides* spp. (**C**), *Eimeria bovis*. (**D**), *Trichostrongylus* spp. (**E**); *Eimeria zuernii* (**F**).

**Figure 5 vetsci-09-00537-f005:**
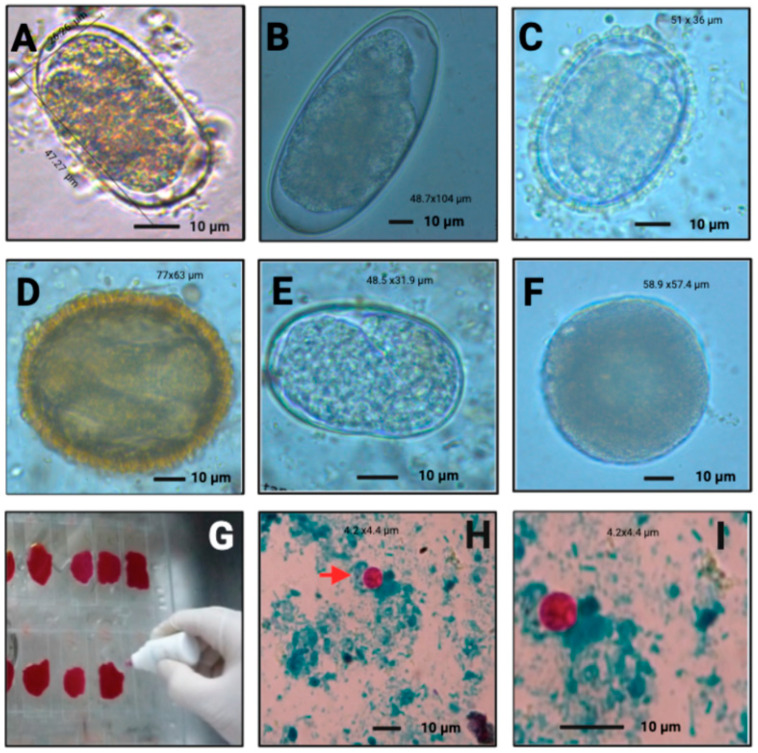
Eggs and cysts found in *T ornatus*. *Ancylostoma* spp. (**A**), *Strongylus* spp. (**B**); *Ascaris* spp. (**C**), *Baylisascaris* spp. (**D**), *Strongyloides* spp. (**E**), *B. coli* (**F**). Smear with carbon fuchsine exhibiting acid-fast staining *Ziehl*–*Neelsen* (**G**), *Cryptosporidium* oocysts 40× (**H**) and 100× (**I**).

**Figure 6 vetsci-09-00537-f006:**
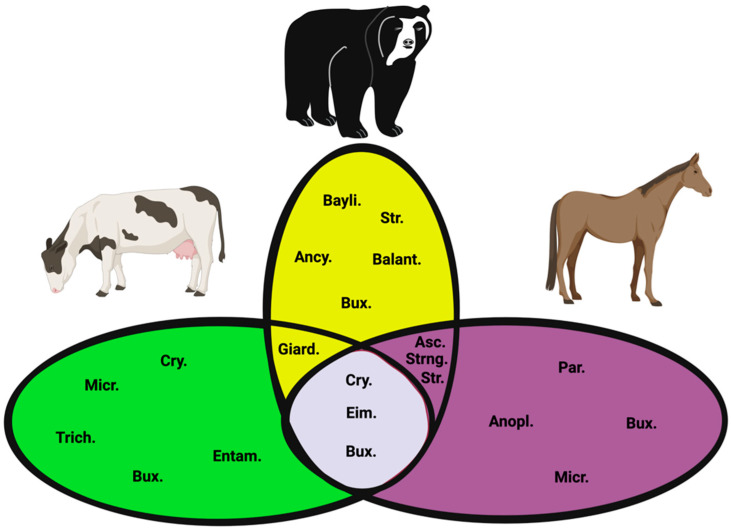
Endoparasites between domestic animals and *T. ornatus*. Eim: *Eimeria* spp., Cry: *Cryptosporidium* spp., Giard: *Giardia* spp., Micro: *Microsporidium* spp., Trich: *Trichostrongylus* spp., Entam: *Entamoeba* spp., Str: Strongyle; Taenia: *Taenia* spp., Bux: *Buxtonella sulcata*, Par: *Parascaris equorum*, Strng: *Strongyloide* spp., Asc: *Ascaris* lumbricoides, Ancy: *Ancylostoma* spp., Bayli: *Bailisascaris* spp., Balant: *Balantidium coli*.

**Table 1 vetsci-09-00537-t001:** Prevalence of endoparasites of domestic animals and *T. ornatus* in Tolima, Valle del Cauca, and Cauca (Colombia).

Species	Prevalence Means	Prevalence IC 95%
**Cattle-*Bos tauros***		
*Eimeria* spp.	53.89% (SD ± 4.6%)	41–66%
*Cryptosporidium* spp.	5.36% (SD ± 2.5%)	0.3–11%
*Giardia* spp.	3.57% (SD ± 1.7%)	1.1–8.3%
*Microsporidium* spp.	2.68% (SD ± 1.3%)	1.4–6.8%
*Trichostrongylus* spp.	2.68% (SD ± 1.3%)	1.4–6.8%
*Entamoeba* spp.	1.79% (SD ± 0.9%)	1.6–5.1%
*Fasciola* spp.	1.79% (SD ± 0.86%)	1.6–5.1%
*Buxtonella* spp.	0.89 % (SD ± 0.43)	−1.5–0.03%
** *Equus caballus* **		
*Eimeria* spp.	33.08% (SD ± 2.08%)	16.8–49.4%
*Strongylus* spp.	18.08% (SD ± 2.50%)	4.7–31.4%
*Cryptosporidium* spp.	4.17% (SD ± 2.08%)	2.8–11.1%
*Buxtonella* spp.	4.17% (SD ± 2.50%)	2.8–11.1%
*Taenia* spp.	4.17% (SD ± 5.42%)	2.8–11.1%
*Parascaris equorum*	2.08% (SD ± 2.92%)	2.9–7.0%
*Microsporidium* spp.	2.08% (SD ± 2.92%)	2.9–7.0%
*Strongyloides* spp.	2.08% (SD ± 3.96%)	2.9–7.0%
*Trichonema* spp.	2.08% (SD ± 3.96%)	2.9–7.0%
*Mesocestoides* spp.	2.08% (SD ± 1.46%)	2.9–7.0%
*Dicroelium* spp.	2.08% (SD ± 1.46%)	2.9–7.0%
** *Tremarctos ornatus* **		
*Eimeria* spp.	30.0% (SD ± 7.07%)	18.2–41.8%
*Ascaris* spp.	21.7% (SD ± 5.11%)	11.1–32.3%
*Ancylostoma* spp.	15.0% (SD ± 3.54%)	5.8–24.2%
*Baylisascaris* spp.	13.3% (SD ± 3.14%)	4.6–22.1%
*Cryptosporidium* spp.	10.0% (SD ± 2.36%)	2.3–17.7%
*Balantidium coli*	5.0% (SD ± 1.18%)	0.6–10.6%
*Anaplocephalidae* spp.	3.3% (SD ± 0.79%)	1.3–8.0%
*Acanthamoeba* spp.	1.7% (SD ± 0.39%)	1.6–5.0%
*Dientamoeba* spp.	1.7% (SD ± 0.39%)	1.6–5.0%
*Diphyllobotrium* spp.	1.7% (SD ± 0.39%)	1.6–5.0%
Fluke	1.7% (SD ± 0.39%)	1.6–5.0%
*Giardia* spp.	1.7% (SD ± 0.39%)	1.6–5.0%
*Paramphistomum* spp.	1.7% (SD ± 0.39%)	1.6–5.0%
*Parascaris* spp.	1.7% (SD ± 0.39%)	1.6–5.0%
*Stephanurus* spp.	1.7% (SD ± 0.39%)	1.6–5.0%
*Strongylus* spp.	1.7% (SD ± 0.39%)	1.6–5.0%
*Buxtonella* spp.	1.7% (SD ± 0.39%)	1.6–5.0%

## Data Availability

Not applicable.

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
