# Peer review of "Endoparasites Infecting Domestic Animals and Spectacled Bears (Tremarctos ornatus) in the Rural High Mountains of Colombia"

_vetsci, 2022, doi:10.3390/vetsci9100537_

Round 1

Reviewer 1 Report

This study described the co-infection prevalence of endoparasites in Tremartus ornatus and domestic animals in the rural high mountains of Colombia. Although it provides some epidemiological data of endoparasites in Tremartus ornatus, but the results of this study are preliminary. I feel that the manuscript requires Major revision before it is suitable for publication.

Major comments:

1.     Some species of protozoa should be identified by molecular biology methods, for example: Eimeria, Cryptosporidium.

2.     The keywords need to be modified.

3.     The photos of Cryptosporidium should be provided.

4.     The description of relationship of infected endoparasites between Tremartus ornatus and domestic animals is insufficiency.

Author Response

Author's Reply to the Review Report (Reviewer 1)

Major comments:

  1. Some species of protozoa should be identified by molecular biology methods, for example: Eimeria, Cryptosporidium.

R./ Thank you very much for your comment. We are in the process of developing a complete study using molecular techniques. Samples are stored in glycerol stock at -80oC and we expect to get fundings after the COVID pandemic.

  1. The keywords need to be modified.

R./ We modified the key words: fecal samples, bear parasites; Andean spectacled bear; zoonosis.

  1. The photos of Cryptosporidium should be provided.

R./ We added photos of Cryptosporidum. smear with carbon fuchsine exhibiting acid-fast staining ZiehlNeelsen (G), Cryptosporidium oocysts 40X (H) and 100X (I).

  1. The description of relationship of infected endoparasites between Tremartus ornatus and domestic animals is insufficiency.

R./ Thank you very much for the feedback. We modified the description, as follows: The multiple associations between wild animals, domestic animals and humans are not completely understood [68], and the potential role of hosts for transmission of zoonotic diseases in rural high mountains is not completely explored, as well as other wild animals that can trigger different dynamics. Zoonotic parasites such as Uncinaria spp., Strongyloides spp., Baylisascaris spp. and Cryptosporidium spp. are present in T. ornatus and domestic animals. This environment  can cause potential larval migrans, skin problems as well as enteric human, domestic and wild infections. Previous reports in humans at high rural mountains by Peña-Quistial shows that Toxocara canis and Toxocara cati had a prevalence of 24% and 44%[14] indicating that these parasites might be circulating in domestic animals that are able to cause larva migrans [14, 68]. In the case of Baylisascaris (Ascarididae family, Ascaridida order, phylum Nematoda), its potential role to infect other animals as well as the agent that can cause larva migrans in humans and animals requires further research.

Reviewer 2 Report

Endoparasites infecting domestic animals and spectacled bears (Tremarctos ornatus) in rural high mountains of Colombia is an interesting study about the presence of parasites also of zoonotic potential in wild endangered species and domestic animals in Colombian regions and in my opinion it deserves to be published under revisions.

Abstract should be improved in terms of content, as it should not include a list of specific results but a summary of the study, with a brief introduction, most relevant results and their discussion. Please prepare a new abstract.

Major comments

Regarding paragraph 2.4 how do you performed the sampling from T. ornatus? it is not detailed.

Please change lines 190-200 as the they are a repetition of the Table 1. Maybe you can report only some of these results, the highest for each host species for example or according to other criteria of interest, but please do not list all results.

line 202-207: you reported features of stool samples, but it is not completely clear how did you use then this information. Are these info related to the type of the study (cross-sectional study seeks to assess the association between the disease or health-related traits and other variables of interest in a specific population and time??). it is completely unclear. Moreover, it is not clear if the endangered species T. ornatus showed some clinical symptoms.

Minor comments

line 65 and 66 Baylisascaris venezuelensis and Baylisascaris transfuga in italics all the species names and along the manuscript (text and figure captions)

line 73 We hope to contribute information regarding T. ornatus’ ecology and parasite niche relations. Please remove we hope and rephrase

LINE 198 were prevalent should be changed with were present or rephrase

line 207 please move the citation to Table 1 in the proper line (maybe 201) and explain the table caption IC.

line 369 typo error acaridae family

line 371-373 is speculative, please rephrase.

Author Response

Author's Reply to the Review Report (Reviewer 2)

Endoparasites infecting domestic animals and spectacled bears (Tremarctos ornatus) in rural high mountains of Colombia is an interesting study about the presence of parasites also of zoonotic potential in wild endangered species and domestic animals in Colombian regions and in my opinion, it deserves to be published under revisions.

Abstract should be improved in terms of content, as it should not include a list of specific results but a summary of the study, with a brief introduction, most relevant results and their discussion. Please prepare a new abstract. 

R/. We modified the abstract, as follows: This research described the co-infection prevalence of endoparasites in Tremartus ornatus and domestic animals in the rural high mountains of Colombia by copro parasitological examination. Some parasites have a zoonotic potential in wild endangered species and domestic animals in Colombian regions. T. ornatus had a notable infection with Eimeria spp, Ascaris spp, Ancylostoma spp, and Baylisascaris spp. Cryptosporidium spp, Balantidium coli, Anoplocephala spp, and Acanthamoeba spp. In B. taurus, Eimeria spp. is coinfecting with Cryptosporidium spp. (6.6%) and represents 18% of the total parasitism. In E. caballus and B. taurus. Eimeria spp. coinfecting (34.7%), with the Strongylus spp. (21.9%-25%). In T. ornatus, Eimeria spp. is coinfecting with Ancylostoma spp. (36.2%), Cryptosporidium spp., Ascaris spp., Baylisascaris spp., and B. coli.

Major comments

Regarding paragraph 2.4 how do you performed the sampling from T. ornatus? It is not detailed.

R/. Stool samples (10 grams) were obtained from domestic animals, horses, and cattle on the border of the reserve forest, directly from the rectum. Between July 13th and December 6th, 2021, we collected fresh feces in the morning (6-12 hours old), which were identified with the aid of an experienced park ranger. Fresh samples were recognized by their brown or green color. Saline wet mounts were made by mixing approximately 2 mg of stool with a drop of physiological saline on a microscope glass slide and placing a coverslip over the stool suspension. Samples were also analyzed using iodine wet mounts and microscopically examined with the aforementioned method. The wet mounts were studied microscopically with a low power objective (10x) and then switching to a high power one (40x). Each stool sample was screened by an experienced microscopist before reporting negative results. Additionally, the Zieh Nielsen technique was employed using 10g of fuchsine di-luted in 100ml of Ethanol, and a 5% of phenol solution (5ml of phenol and 95ml of water). Then, 10ml of basic fuchsine is filtered and 100ml of phenol solution is added in order to form the mother solution. The excess of alcohol was removed with tap water, discolored with 7% H2SO4 until the plate was pale pink, forming a sulphuric acid solution (7% H2S04, 7ml of Sulphuric acid mixed with 93ml of Ethanol). The excess of colorant was also removed with tap water, then we added methylene blue or malachite green, spreading it for 3 minutes. 10g of methylene blue is diluted in 95% Ethanol then 30ml is filtered from the 100ml of the mother solution; afterwards, 70ml of water is added. The malachite green solution is conformed of 5g malachite green diluted in 10% Ethanol, 100ml). The excess of colorant is eliminaed with tap water and left to dry in order to visualize the plate with immersion oil, using the 100x objective. The parasite analysis was performed by direct microscopic examination using a ZEISS AxioCam ICc 1 microscope, with flotation using the Sheather technique and sedimentation methods, as well as fixation and coloring techniques of Zieh Nielsen [10, 14]. Samples were stored at -20oC for future molecular studies.

Please change lines 190-200 as the they are a repetition of the Table 1. Maybe you can report only some of these results, the highest for each host species for example or according to other criteria of interest, but please do not list all results.

  1. Thank you, we corrected it and limited the report to some of the results.

line 202-207: you reported features of stool samples, but it is not completely clear how did you use then this information. Are these info related to the type of the study (cross-sectional study seeks to assess the association between the disease or health-related traits and other variables of interest in a specific population and time??). it is completely unclear. Moreover, it is not clear if the endangered species T. ornatus showed some clinical symptoms.

R: Thank you for your comment. We emphasized in the text that neither T. ornatus nor domestic animals did show clinical symptoms and it was not part of the cross- sectional study to assess disease of health-related traits and other variables.

Minor comments

line 65 and 66 Baylisascaris venezuelensis and Baylisascaris transfuga in italics all the species names and along the manuscript (text and figure captions)

Thank you. Line 65 and 66 was already changed in italics.

line 73 We hope to contribute information regarding T. ornatus’ ecology and parasite niche relations. Please remove we hope and rephrase

  1. We aim to contribute information about T. ornatus’ ecology and parasite niche relations.

LINE 198 were prevalent should be changed with were present or rephrase

R./ Thank you, it has been changed.

line 207 please move the citation to Table 1 in the proper line (maybe 201) and explain the table caption IC. 

R./ Table 1. was in the first paragraph.

line 369 typo error acaridae family

  1. It is a mistake, we corrected it to “Ascarididae family”.

line 371-373 is speculative, please rephrase.

R./ Thank you, we already rephrase it to “Baylisascaris spp. has been previously published as B. venezuelensis, since it has already been characterized in T. ornatus using molecular techniques and compared with Baylisascaris transfuga, which has shown a 52.9% prevalence in brown bears [64, 65]”.

Kind regards

Round 2

Reviewer 1 Report

I am satisfied with the author's revision.